# N6-methyladenosine helps *Apostichopus japonicus* resist *Vibrio splendidus* infection by targeting coelomocyte autophagy via the AjULK-AjYTHDF/AjEEF-1α axis

Jiqing Liu[1], Yina Shao[1], Dongdong Li[1] & Chenghua Li ⓘ [1,2✉]

N6-Methyladenosine (m6A) modification is one of the most abundant post-transcriptional modifications that can mediate autophagy in various pathological processes. However, the functional role of m6A in autophagy regulation is not well-documented during *Vibrio splendidus* infection of *Apostichopus japonicus*. In this study, the inhibition of m6A level by knockdown of methyltransferase-like 3 (AjMETTL3) significantly decreased *V. splendidus*-induced coelomocyte autophagy and led to an increase in the intracellular *V. splendidus* burden. In this condition, *Unc-51-like kinase 1* (*AjULK*) displayed the highest differential expression of m6A level. Moreover, knockdown of *AjULK* can reverse the *V. splendidus*-mediated autophagy in the condition of AjMETTL3 overexpression. Furthermore, knockdown of *AjMETTL3* did not change the *AjULK* mRNA transcript levels but instead decreased protein levels. Additionally, YTH domain-containing family protein (AjYTHDF) was identified as a reader protein of *AjULK* and promoted *AjULK* expression in an m6A-dependent manner. Furthermore, the AjYTHDF-mediated *AjULK* expression depended on its interaction with translation elongation factor 1-alpha (AjEEF-1α). Altogether, our findings suggest that m6A is involved in resisting *V. splendidus* infection via facilitating coelomocyte autophagy in AjULK-AjYTHDF/AjEEF-1α-dependent manner, which provides a theoretical basis for disease prevention and therapy in *A. japonicus*.

[1] State Key Laboratory for Quality and Safety of Agro-products, Ningbo University, Ningbo, P. R. China. [2] Laboratory for Marine Fisheries Science and Food Production Processes, Qingdao National Laboratory for Marine Science and Technology, Qingdao, P. R. China. ✉email: lichenghua@nbu.edu.cn

Autophagy refers to a cellular self-digestion process in which wrapped cytoplasm components, including invading pathogens, damaged organelles, or proteins, are encapsulated by a double membrane and subsequently degraded by lysosomes[1]. The main function of autophagy is to maintain cell survival when cells are threatened by stressful death, and it has become an evolutionarily conserved mechanism for eukaryotic cells to maintain homeostasis and achieve self-renewal[2–4]. Autophagy is also involved in innate immune defenses against invading pathogens, such as *Shigella flexneri*, *Mycobacterium tuberculosis*, *Toxoplasma gondii*, and *Listeria monocytogenes*[5–8]. The type of autophagy that selectively transported microorganisms to lysosomes is termed xenophagy[9]. In mammals (and perhaps other metazoan organisms), the role of autophagy in antimicrobial defense probably extends beyond the direct elimination of pathogens[10]. An increasing number of studies have shown the great contribution of post-transcriptional regulation on RNA to autophagy signaling pathway[11,12].

N6-Methyladenosine (m6A), as one of the most abundant means of post-transcriptional regulation, plays an important role in a variety of physiological and pathological processes[13,14]. m6A mainly occurs in RRACH sequences and is orchestrated mainly by several m6A regulators, including writers, erasers, and readers[15,16]. Methyltransferase like 3 (METTL3), METTL14, and Wilms' tumor 1-associating protein are mainly "writers" for catalyzing the formation of m6A modification[17]. For maintaining m6A modification in a dynamic balance, fat mass and obesity-associated gene (FTO) and AlkB homologue 5, RNA demethylase (ALKBH5) can remove the modification[18]. The reader proteins mainly include YTH m6A RNA-binding protein family, insulin-like growth factor 2 mRNA-binding protein family, and heterogeneous nuclear ribonucleoprotein C, which can recognize modifications to influence mRNA stability, translation, subcellular localization, and alternative splicing to perform different biological functions[19]. YTH m6A RNA-binding protein family is the most functional reader protein[20].

The relationship between autophagy and m6A was first described in *Homo sapiens* in 2018. The m6A distribution within *ULK* transcripts can be abrogated by FTO, which promotes the production of ULK1 protein, microtubule-associated protein light chain 3 (LC3), and autophagic flux[21]. The way in which m6A regulates autophagy also differs in various organisms or physiological process. FTO depletion-mediated m6A modification of autophagy-related gene-5 (ATG5) and ATG7 transcripts recruit the reader protein YTHDF2, which facilitates degradation and impairs autophagosome formation in *Mus musculus*[22]. Conversely, m6A modification can promote autophagy in certain cases. A typical example is that the depletion of ALKBH5 increases m6A level of *BCL-2* transcript and leads to degradation of *BCL-2* transcript in ovarian cancer cells, which initiates autophagy through disruption of the BCL-2–Beclin1 complex[23].

Autophagy, as a means of innate immunity in *Apostichopus japonicus*, showed an important antibacterial role against *Vibrio splendidus* infection in our previous studies[24,25]. However, the regulatory mechanism behind autophagy remains largely unknown. In the study, we confirmed that a positive correlation exists between m6A modification and autophagy in response to *V. splendidus* infection. Furthermore, *Unc-51-like kinase 1* (*AjULK*), a known autophagy initial factor in *A. japonicus*, was modified by m6A and involved in METTL3 in *A. japonicus* (AjMETTL3)-mediated autophagy. Moreover, a conserved reader protein of YTH domain-containing family protein (AjYTHDF) was identified and confirmed to promote AjULK expression in an m6A-dependent manner. More importantly, we validated that AjYTHDF-mediated AjULK expression depends on its interaction with elongation factor 1-alpha (AjEEF-1α). These findings enrich our knowledge on the role of m6A in innate immunity of

marine invertebrate species and provide insights for disease prevention and therapy of *A. japonicus*.

## Results

**m6A methylation regulates autophagy to resist *V. splendidus* infection**. Some researchers found autophagy can help hosts to defend against pathogenic infection[26]. Our previous studies showed that the autophagy activity and m6A methylation level of coelomocytes significantly increased during *V. splendidus* infection of *A. japonicus*[24,25,27,28]. However, the relationship between autophagy and m6A methylation was not addressed. To reveal whether autophagy is regulated by m6A methylation, we applied siRNA to knockdown the main m6A methyltransferase *AjMETTL3*. qRT-PCR and Western blotting results showed that si-AjMETTL3 can significantly reduce the expression level of AjMETTL3 (Supplementary Fig. 1). First, TEM showed that the number of autophagosomes in si-AjMETTL3 group was significantly reduced compared with that in the si-NC group under *V. splendidus* infection condition (Fig. 1a). Similarly, immunofluorescence analysis showed that AjLC3 green fluorescence and lysosomal red fluorescence in the si-AjMETTL3 group were significantly reduced compared with those in the si-NC group (Fig. 1b, c). Consistently, when AjMETTL3 was knocked down, AjLC3-II/I ratio was down-regulated compared with that in the si-NC group in *V. splendidus*-stimulated conditions. Moreover, under Baf-A1 treatment, the reduction of AjLC3-II/I ratio in the AjMETTL3 knockdown group was further enhanced, which demonstrated that autophagy flux was impaired in AjMETTL3 knockdown condition (Fig. 1d, e). These results collectively indicated that m6A methylation modification served as a positive regulator in *V. splendidus*-induced coelomocytes autophagy. Autophagy in *A. japonicus* can resist *V. splendidus* infection, as has been proven in our previous research[24,25]. To further demonstrate the immunological function of m6A in response to *V. splendidus* infection, we detected bacterial abundance in coelomocytes. The abundance of culturable *V. splendidus* in the si-AjMETTL3 group significantly increased compared with that in the si-NC group (Fig. 1f, g). Collectively, the results proved that m6A played an anti-bacterial role during *V. splendidus* infection of *A. japonicus*.

**Autophagy initiation factor AjULK is the target gene of AjMETTL3**. To further address the molecular mechanism of AjMETTL3-mediated autophagy, we screened the potential target in ATGs. Based on our published results on m6A-seq (https://www.ncbi.nlm.nih.gov/pmc/?term=PRJNA774950%20, PRJNA774950) and ATGs in *A. japonicus*, we screened out 15 ATGs with significantly changed m6A methylation levels after *V. splendidus* infection; *AjULK* was the most changed gene at m6A methylation levels (Fig. 2a)[25,27]. AjULK has been proven as an autophagy initiation factor controlling *V. splendidus*-induced autophagy[24]. Two m6A methylation sites were found in the coding sequence region of *AjULK* transcripts by m6A-seq (Fig. 2b). MeRIP-qPCR also confirmed that the m6A methylation level of AjULK significantly increased in *V. splendidus* infection group compared with that in the untreated group (Fig. 2c). Next, in AjMETTL3 knockdown condition, AjULK significantly decreased at the protein level (Fig. 2e), whereas the mRNA level of *AjULK* showed no change (Fig. 2d). Consistently, the expression of AjULK-downstream genes, including AjBeclin-1 and AjATG13, were also significantly depressed, whereas the protein level of Ajp62 was significantly elevated in this condition (Fig. 2e).

To further explore whether m6A methylation modification on *AjULK* mRNA is crucial for mediating its expression, we constructed wild and mutant luciferase reporter vectors (AjULK-WT and AjULK-Mut, respectively) for the m6A methylation site of *AjULK* (Fig. 2f). Dual-luciferase assay revealed that

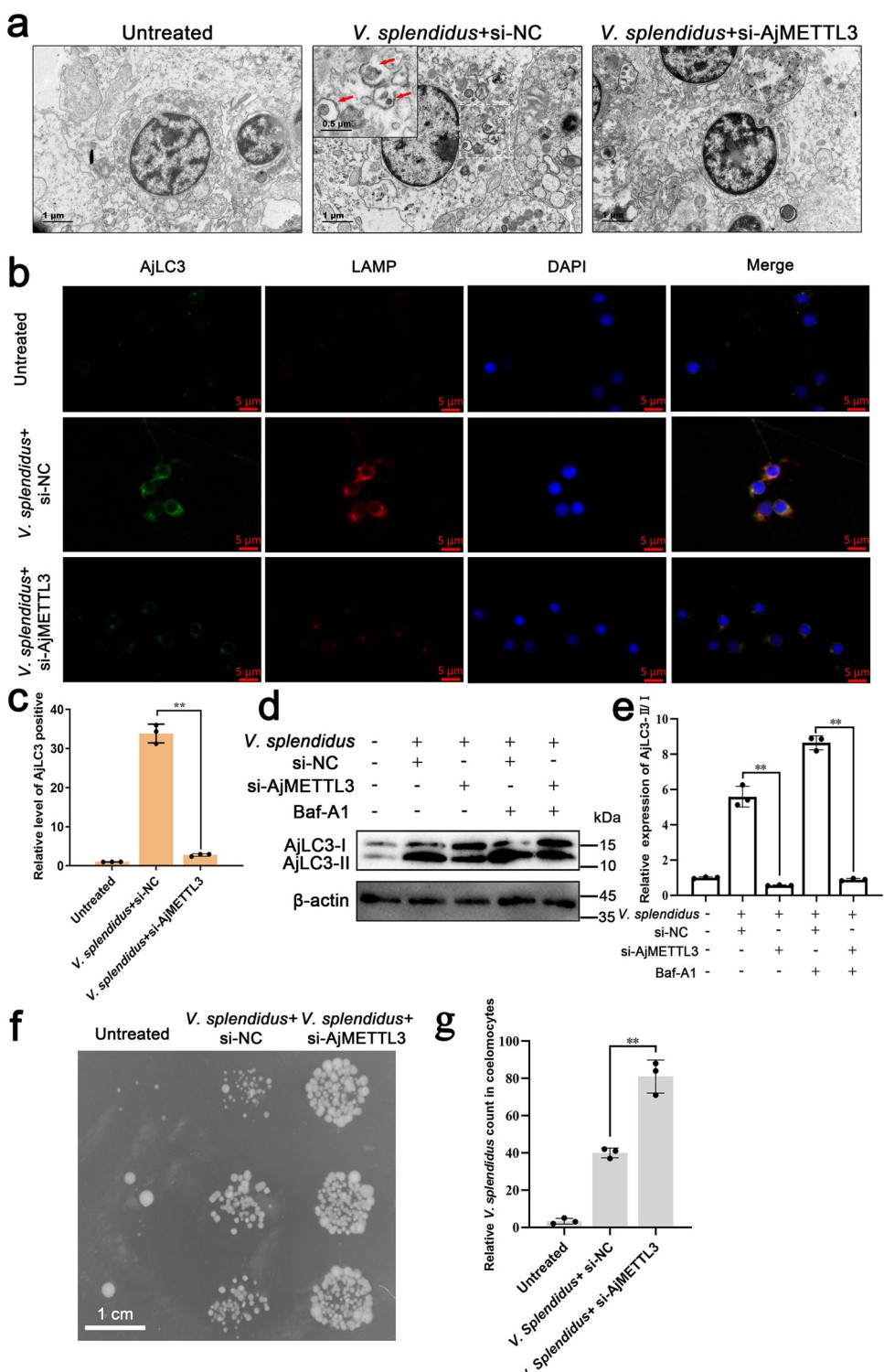

the overexpression of AjMETLL3 significantly up-regulated the luciferase reporter activity in the AjULK-WT-2 group, whereas the luciferase reporter activities of AjULK-WT-1, AjULK-MUT-1, and AjULK-MUT-2 were not changed (Fig. 2g). These results indicate that m6A promoted the AjULK expression via the second m6A methylation site.

**m6A regulates *V. splendidus*-induced autophagy through AjULK.** As the target of m6A modification, *AjULK* was speculated to be involved in m6A-mediated autophagy. To confirm this

hypothesis, we assayed autophagy in different conditions. After injecting *AjMETTL3* mRNA in coelomocytes, the AjMETTL3 mRNA and protein levels in the OE-AjMETTL3 group increased compared with the OE-NC group ($p < 0.01$), respectively (Fig. 3a, b). Meanwhile, the protein level of AjULK in the OE-AjMETTL3 group was also up-regulated compared with that in the OE-NC group, in which the mRNA level of *AjULK* exhibited no change (Fig. 3a, b). In this condition, *V. splendidus* infection can significantly increase the number of autophagosomes in the OE-AjMETTL3 group compared with the OE-NC group, as shown by TEM observation (Fig. 3c). However, knockdown of AjULK can reverse the

**Fig. 1 m6A methylation facilitates coelomocyte autophagy and resists V. splendidus infection. a** After *A. japonicus* were transfected with si-NC or si-AjMETTL3 for 24 h and then infected by $1 \times 10^7$ CFU/mL *V. splendidus* for 24 h, TEM showed that the increase in autophagosomes in the si-NC + *V. splendidus* group significantly decreased in the si-AjMETTL3 + *V. splendidus* group. Red arrows indicate autophagosomes. Scale bar = 1 μm. **b** After *A. japonicus* were transfected with si-NC or si-AjMETTL3 for 24 h and then infected by $1 \times 10^7$ CFU/mL *V. splendidus* for 24 h, immunofluorescence analysis revealed that the increased fluorescence intensity of AjLC3 and lysosomes in the si-NC + *V. splendidus* group decreased greatly in the si-AjMETTL3 + *V. splendidus* group. Blue, green, and red signals represent nuclear, AjLC3, and lysosomes, respectively. Scale bar = 5 μm. **c** Relative AjLC3 positivity in 1000 cells from each indicated sample was determined. The data are presented as means ± standard deviations (SDs) (*n* = 3) relative to the untreated group and shown in bar graphs. Asterisks indicate significant differences between groups: *$p < 0.05$ and **$p < 0.01$ by one-way ANOVA. **d** After *A. japonicus* were transfected with si-NC or si-AjMETTL3 for 24 h and then infected by $1 \times 10^7$ CFU/mL *V. splendidus* for 24 h, Western blot analysis revealed that the upregulated AjLC3-II/I ratio in the si-NC + *V. splendidus* group was reduced in the si-AjMETTL3 + *V. splendidus* group. Further, in *A. japonicus* treated by 10 nm Baf-A1, the reduction of AjLC3-II/I ratio in the AjMETTL3 knockdown group compared with the si-NC group in *V. splendidus*-stimulated conditions was further enhanced. The bands presented were from same blot membrane and same quantity of protein (50 μg) were loaded into each lane. **e** Protein band density was calculated using ImageJ software. The data are presented as means ± SDs (*n* = 3) relative to the untreated group and shown in bar graphs. Asterisks indicate significant differences between groups: *$p < 0.05$ and **$p < 0.01$ by one-way ANOVA. **f, g** After *A. japonicus* were transfected with si-NC or si-AjMETTL3 for 24 h and then injected by 100 μl *V. splendidus* ($10^6$ CFU/mL) for another 24 h, plate-counting method revealed the bacterial load in coelomocytes; the abundance of *V. splendidus* in the si-AjMETTL3 + *V. splendidus* group significantly increased compared with the si-NC + *V. splendidus* group. Scale bar = 1 cm. The data are presented as means ± SDs (*n* = 3) relative to the untreated group and shown in bar graphs. Asterisks indicate significant differences between groups: *$p < 0.05$ and **$p < 0.01$ by one-way ANOVA. Error bars represent SDs.

increased autophagosomes in AjMETTL3-overexpressing coelomocytes (Fig. 3c). Consistently, immunofluorescence observation showed that the overexpression of AjMETTL3 also increased the AjLC3 fluorescence signal, whereas the increased AjLC3 fluorescence signal in AjMETTL3-overexpressing coelomocytes were reversed by the knockdown of AjULK (Fig. 3d, e). Similar to these results, AjLC3-II/I ratio in AjMETTL3-overexpressing coelomocytes was upregulated compared with that in the OE-NC group. Nonetheless, the loss of AjULK also recovered the augmented AjLC3-II/I ratio in AjMETTL3-overexpressing coelomocytes, which revealed that knockdown of AjULK can block AjMETTL3-regulated autophagy flux (Fig. 3, g). In addition, overexpression of AjMETTL3 can decrease the intracellular *V. splendidus* burden in coelomocytes compared with the OE-NC group. Knockdown of AjULK in this condition can reverse the intracellular *V. splendidus* number in coelomocytes (Fig. 3h, i). These results indicate that AjMETTL3 mediates autophagy to resist *V. splendidus* infection through AjULK in *A. japonicus*.

**AjYTHDF is a reader protein that mediates AjULK expression.** m6A methylation influences mRNA stability and translation, which are mediated by specific m6A-binding proteins[29,30]. YTHDF1 recognizes and promotes the translation of m6A-modified mRNA[31]. In *A. japonicus* transcriptome, only one AjYTHDF member was found, and it showed the highest homology to YTHDF1 in the phylogenetic tree (Supplementary Fig. 2). To confirm that AjYTHDF was a reader protein in m6A-mediated AjULK expression, we performed expression analysis and functional validation. First, qRT-PCR analysis showed that the expression pattern of *AjYTHDF* was up-regulated in *V. splendidus* infection process, which was similar to the expression pattern of *AjULK* (Fig. 4a). Second, *AjYTHDF* siRNA transfection significantly decreased the *AjYTHDF* mRNA and protein expressions, respectively (*p* < 0.01) (Fig. 4b, c). In this condition, the protein level of AjULK significantly decreased (Fig. 4c), whereas the mRNA level of *AjULK* showed no change (Fig. 4b). Meanwhile, the expressions of AjBeclin-1 and AjATG13 were also significantly downregulated, whereas that of Ajp62 was significantly increased (Fig. 4c). Furthermore, RNA pull-down assay was used to confirm whether m6A-modified *AjULK* transcripts can specifically interact with AjYTHDF. The results indicated that m6A-labeled AjULK probes can enrich AjYTHDF proteins rather than the Mut probes of AjULK (Fig. 4d). Next, RIP-qPCR assay was further used to investigate whether AjYTHDF can recognize and bind m6A-modified *AjULK* mRNA. Our data showed that the enrichment of

*AjULK* mRNA in complexes precipitated with the antibody against AjYTHDF compared with those with untreated IgG (Fig. 4e). When we increased the m6A level of *AjULK* by AjMETTL3 overexpression, the level of *AjULK* mRNA enriched by AjYTHDF antibody was further increased. Conversely, the down-regulated m6A level of *AjULK* by AjMETTL3 knockdown led to the reduced interaction between AjYTHDF and *AjULK* mRNA (Fig. 4e). In addition, RNA EMSA confirmed that AjYTHDF protein can specifically bind with the m6A-modified *AjULK* rather than Mut *AjULK* in vitro. Supershift experiments indicated that AjYTHDF specifically binds to this sequence. When the concentration of AjYTHDF protein increased, the concentration of the *AjULK*/AjYTHDF mixture also increased. Mutation of the m6A site of *AjULK* abolished the interaction (Fig. 4f).

In the further investigation of AjYTHDF as a reader protein of *AjULK* that mediates autophagy, immunofluorescence observation indicated that knockdown of AjYTHDF abrogated the increased AjLC3 fluorescence signal by overexpression of AjMETTL3 under *V. splendidus*-stimulated condition (Fig. 4g, h). Consistently, Western blot showed that AjULK and autophagy flux in the OE-AjMETTL3 + si-AjYTHDF group were reduced compared with those in the OE-AjMETTL3 group under *V. splendidus*-stimulated condition (Fig. 4I, j). The results demonstrated that AjYTHDF, as a reader protein of *AjULK*, promotes AjULK expression to mediate autophagy.

**AjYTHDF regulates AjULK expression in an m6A-dependent manner.** In mammal, YTHDF1 binds m6A sites through its m6A-binding pockets in YTH domain, in which K395 and Y397 are the key sites for YTHDF1 bound with m6A-modified mRNA[32]. In AjYTHDF, the K464 and Y466 sites were conserved to K395 and Y397, respectively (Supplementary Fig. 3). To investigate whether AjYTHDF regulates AjULK expression in an m6A-dependent manner, we introduced two points mutations (K464 and Y466) to the YTH domain of AjYTHDF (AjYTHDF-Mut) (Fig. 5a). Then, the mRNAs transcribed using AjYTHDF wild-type (AjYTHDF-WT) or AjYTHDF-Mut as template were injected to *A. japonicus* to overexpress the different types of AjYTHDFs. After the overexpression of AjYTHDFs, the expression of AjULK was upregulated in the AjYTHDF-WT group, whereas it was unchanged in the AjYTHDF-Mut group (Fig. 5b). Accordingly, RIP-qPCR assay revealed that the enrichment of *AjULK* mRNA by AjYTHDF did not increase in the AjYTHDF-Mut group compared with the untreated group (Fig. 5c). Furthermore, the dual-luciferase reporter gene system showed that

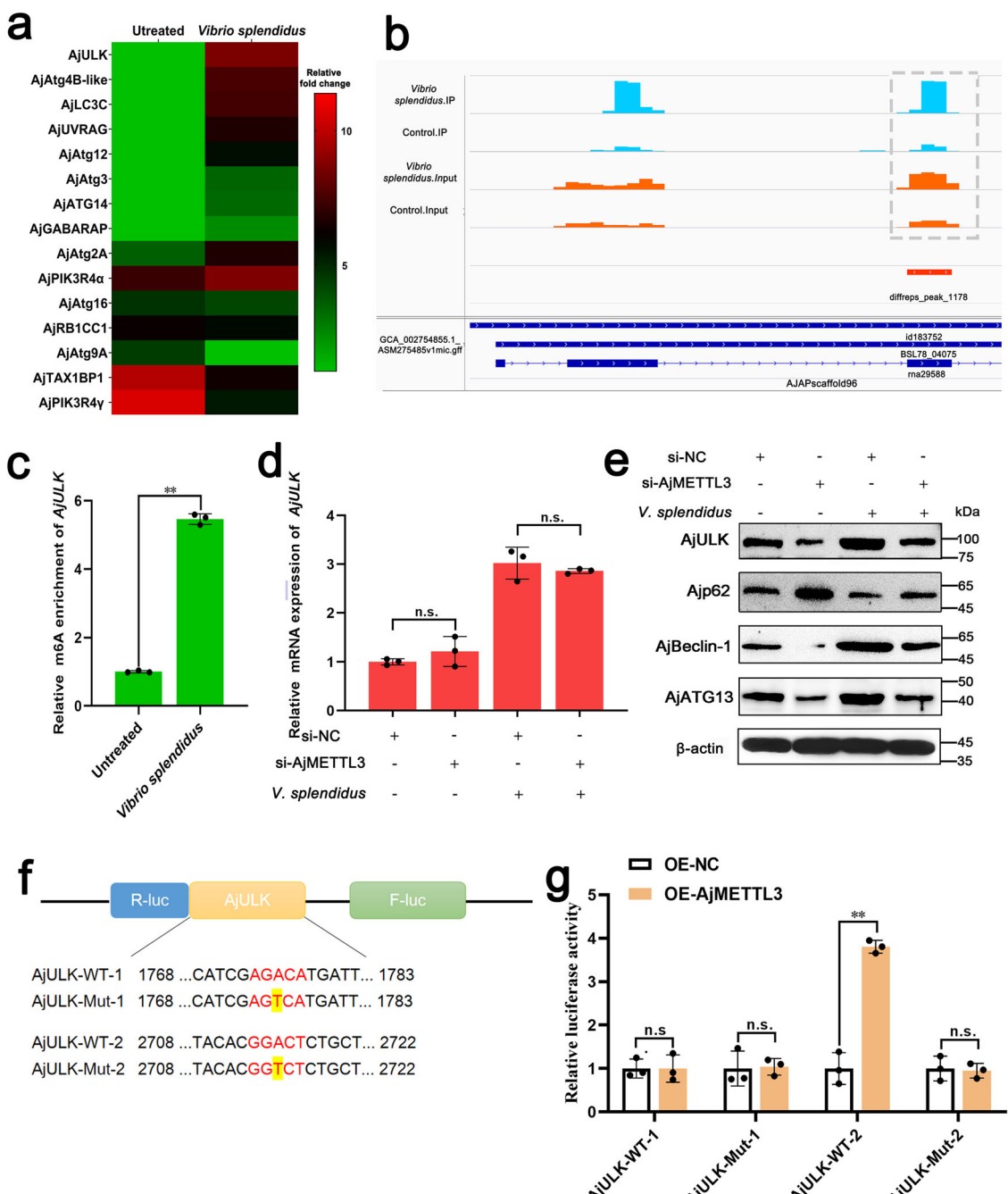

**Fig. 2 Autophagy initiation factor *AjULK* is the target gene of AjMETTL3. a** Heat map showing the m6A methylation levels of ATGs in response to *V. splendidus* infection. The color ranges from green to red, indicating m6A level from low to high. **b** Integrative genomics viewer plots of m6A peaks in *AjULK* mRNAs. Blue boxes represent exons, and blue lines denote introns. **c** MeRIP-qPCR analysis of the m6A level of *AjULK* in response to *V. splendidus* infection. The data are presented as means ± SDs ($n = 3$) relative to the untreated group and shown in bar graphs. Asterisks indicate significant differences between groups: *$p < 0.05$ and **$p < 0.01$ by Student's t-test. **d, e** After *A. japonicus* were infected or uninfected by $1 \times 10^7$ CFU/mL *V. splendidus* for 24 h in AjMETTL3 knockdown condition, qRT-PCR and Western blot analysis of the mRNA and protein levels of *AjULK* and autophagy-related proteins in coelomocytes, respectively, were conducted. The protein levels of AjULK in the si-AjMETTL3 group were significantly down-regulated compared with those of the proteins in the si-NC group. The mRNA level of *AjULK* in the si-AjMETTL3 group did not change compared with that in the si-NC group. Under *V. splendidus* infection conditions, a similar phenomenon was also observed. The data are presented as means ± SDs ($n = 3$) relative to the si-NC group and shown in bar graphs. Asterisks indicate significant differences between groups: *$p < 0.05$ and **$p < 0.01$ by one-way ANOVA. n.s., not significant ($p > 0.05$). The bands presented were from five blot membranes and same quantity of protein (50 μg) were loaded into each lane. **f** Schematic of dual-luciferase reporter constructs. **g** Relative luciferase activity of AjULK-WT or AjULK-Mut luciferase reporter in AjMETTL3-overexpressing cells. Overexpression of AjMETLL3 significantly upregulated the luciferase reporter activity in the AjULK-WT-2 group, whereas the luciferase reporter activity of AjULK-WT-1, AjULK-MUT-1, and AjULK-MUT-2 showed no change. Renilla luciferase activity was measured and normalized to firefly luciferase activity. The data are presented as means ± SDs ($n = 3$) relative to the OE-NC group and shown in bar graphs. Asterisks indicate significant differences between two groups: *$p < 0.05$ and **$p < 0.01$ by two-way ANOVA. n.s., not significant ($p > 0.05$). Error bars represent SDs.

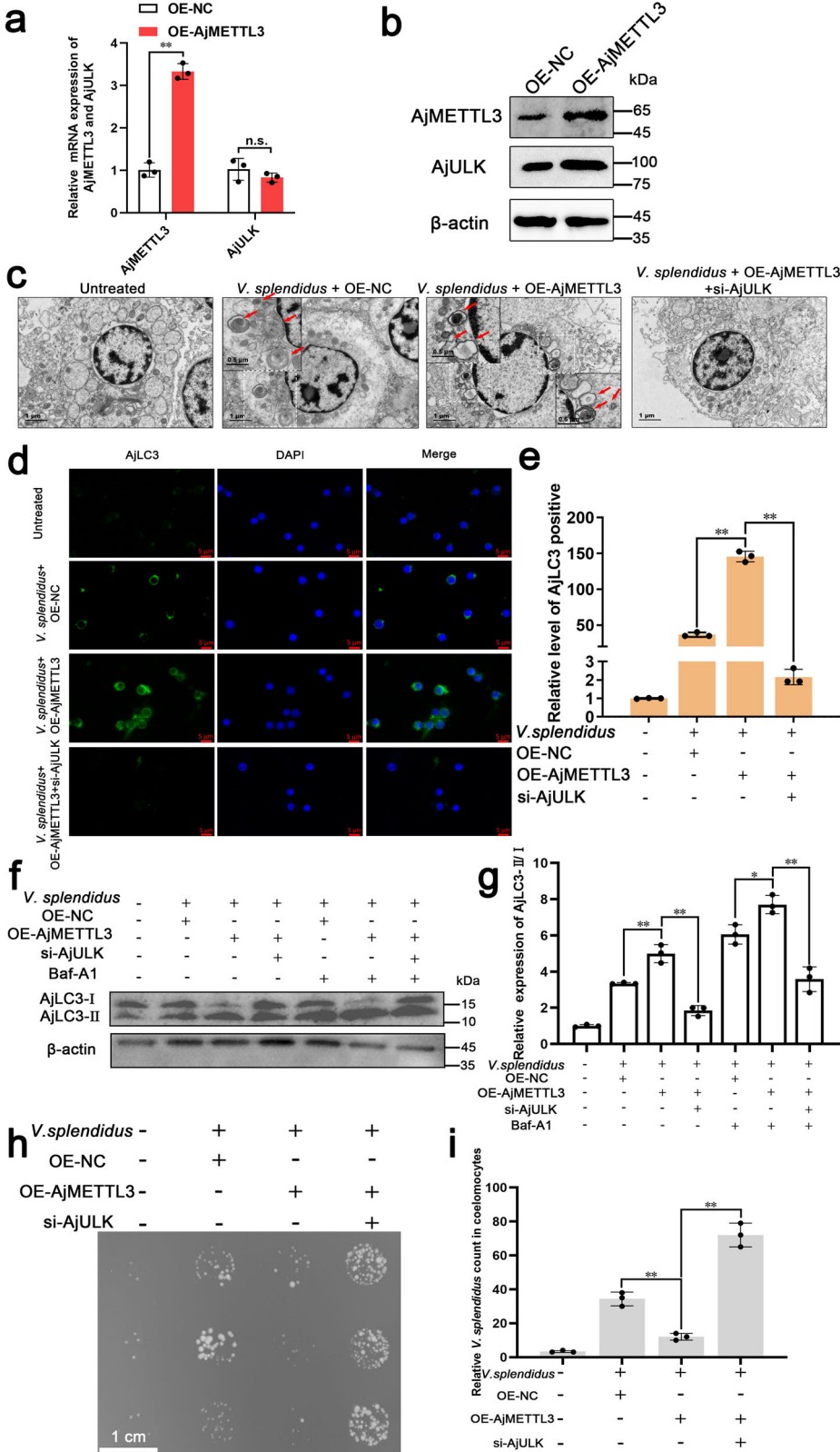

the overexpression of AjYTHDF can upregulate the luciferase activity of the AjULK-WT-2 group, whereas the AjULK-Mut-2 group had no response to the overexpression of AjYTHDF (Fig. 5d). In addition, the overexpression of AjYTHDF cannot upregulate AjULK expression in coelomocytes lacking AjMETTL3 (Fig. 5e). All the results suggest that AjYTHDF regulates AjULK expression in an m6A-dependent manner.

**AjYTHDF-mediated AjULK expression depends on the interaction with AjEEF-1α.** To further investigate the mechanism of AjYTHDF in promoting the expression of AjULK transcripts, we performed AjYTHDF-GST pull-down assay on coelomocytes and characterized the results by MS. Interestingly, we found AjEEF-1α was enriched among the proteins (Fig. 6a). Next, we further used the purified protein of AjYTHDF and AjEEF-1α to perform GST

**Fig. 3 m6A regulates *V. splendidus*-induced autophagy through AjULK. a**, **b** qRT-PCR and Western blot analysis of the overexpression efficiency of AjMETTL3 and expression of AjULK in coelomocytes after 24 h postinjection with 300 µg AjMETTL3 mRNA. The *AjMETTL3* mRNA and protein levels in the OE-AjMETTL3 group increased compared with the OE-NC group, respectively. Meanwhile, the protein level of AjULK in the OE-AjMETTL3 group was also up-regulated compared with that in the OE-NC group, in which the mRNA level of *AjULK* exhibited no change. The data are presented as means ± SDs ($n = 3$) relative to the OE-NC group and shown in bar graphs. Asterisks indicate significant differences between groups: *$p < 0.05$ and **$p < 0.01$ by Student's t-test. n.s., not significant ($p > 0.05$). The bands presented were from three blot membranes and same quantity of protein (50 µg) were loaded into each lane. **c** After simultaneous overexpression of AjMETTL3 in *A. japonicus*, knockdown of AjULK, and infection by $1 \times 10^7$ CFU/mL *V. splendidus* for 24 h, TEM analysis revealed autophagosomes in coelomocytes. The number of autophagosomes in the OE-AjMETTL3 group significantly increased compared with the OE-NC group. In addition, AjULK knockdown can reverse the increased autophagosomes by AjMETTL3 overexpression. Red arrows indicate autophagosomes. Scale bar = 1 µm. **d** After simultaneous overexpression of AjMETTL3 in *A. japonicus*, AjULK knockdown, and infection by $1 \times 10^7$ CFU/mL *V. splendidus* for 24 h, immunofluorescence analysis of coelomocytes was conducted and revealed that AjULK knockdown can reduce the increased fluorescence intensity of AjLC3 by AjMETTL3 overexpression. Blue, green, and red signals represent nuclear, AjLC3, and lysosomes, respectively. Scale bar = 5 µm. **e** Relative AjLC3 positivity in 1000 cells from each indicated sample was determined. The data are presented as means ± SDs ($n = 3$) relative to the untreated group and shown in bar graphs. Asterisks indicate significant differences between groups: *$p < 0.05$ and **$p < 0.01$ by one-way ANOVA. **f** After simultaneous overexpression of AjMETTL3 in *A. japonicus*, knockdown of AjULK, and infection by $1 \times 10^7$ CFU/mL *V. splendidus* for 24 h, Western blot analysis was used to determine AjLC3-II/I ratio, and it showed that the loss of AjULK also recovered the augmented AjLC3-II/I ratio in AjMETTL3-overexpressing coelomocytes. The bands presented were from same blot membrane and same quantity of protein (50 µg) were loaded into each lane. **g** Protein band density was calculated using ImageJ software. Data are presented as means ± SDs ($n = 3$) relative to the untreated group and shown in bar graphs. Asterisks indicate significant differences between groups: *$p < 0.05$ and **$p < 0.01$ by one-way ANOVA. **h**, **i** After simultaneous overexpression of AjMETTL3 in *A. japonicus*, knockdown of AjULK, injection with 100 µl *V. splendidus* ($10^6$ CFU/mL) for 24 h, plate-counting method analysis revealed the bacterial load in coelomocytes. In addition, the decreased *V. splendidus* in AjMETTL3-overexpressing *A. japonicus* was reversed by the knockdown of AjULK. Scale bar = 1 cm. The data are presented as means ± SDs ($n = 3$) relative to the untreated group and shown in bar graphs. Asterisks indicate significant differences between groups: *$p < 0.05$ and **$p < 0.01$ by one-way ANOVA. Error bars represent SDs.

pull-down assay. The results showed that GST-AjYTHDF can enrich His-AjEEF-1α, which confirmed that the two proteins directly interacted (Supplementary Fig. 4). Moreover, co-IP and reverse Co-IP experiments confirmed that AjYTHDF interacted with AjEEF-1α in a dose-dependent manner (Fig. 6b, c). In the investigation determining whether AjEEF-1α is necessary for m6A-mediated AjULK expression, knocked down AjEEF-1α eliminated the AjULK increased expression in AjMETTL3 overexpression condition (Fig. 6f, g). Furthermore, the knockdown of AjEEF-1α in the OE-AjMETTL3 group reduced the signal of AjLC3 compared with that in the OE-AjMETTL3 group (Fig. 6d, e). Consistently, autophagy flux in the si-AjEEF-1α + OE-AjMETTL3 group decreased compared with that in the OE-AjMETTL3 group (Fig. 6f, g). These results indicate that the combination of AjYTHDF with AjEEF-1α is essential to promote the expression of m6A-modified *AjULK*.

## Discussion

m6A, as one of the most prevalent RNA modifications, plays an important role in the regulation of the stability and translation of mRNAs and is involved in various biological processes[33–35]. m6A RNA methylation can alter the expression of ATGs and influence the autophagy function[36,37]. In our previous studies, abnormal m6A level and autophagy were found in *A. japonicus* infected with *V. splendidus*[24,25,27]. However, whether m6A can regulate autophagy to resist *V. splendidus* infection is unclear. In this study, knockdown of AjMETTL3 can significantly inhibit autophagy activity and lead to an increased intracellular *V. splendidus* abundance in *A. japonicus* during *V. splendidus* infection. Further studies revealed that AjMETTL3 regulated autophagy by targeting AjULK expression. As for the mechanism, m6A-modified *AjULK* was recognized by AjYTHDF, and AjYTHDF regulated AjULK expression in an m6A-dependent manner. In addition, AjYTHDF interaction with AjEEF-1α was essential to promote AjULK expression (Fig. 7). Overall, our current study filled in the gap that m6A regulates autophagy against pathogen infection in *A. japonicus*.

Autophagy is an intracellular clearance mechanism that can be regulated through numerous ways[38–40]. An increasing number of studies indicated that m6A plays a vital role in autophagy

regulation, which can promote or inhibit autophagy in a number of physiological and pathological processes[41–44]. In our study, m6A played a positively regulating role in *V. splendidus*-induced autophagy in *A. japonicus*. In most cases, m6A modification mediates autophagy mainly via affecting the expression of ATGs[45,46]. m6A can control autophagy by targeting *ATG5* and *ATG7* in adipogenesis[22]. METTL3-mediated m6A modification attenuates *ATG7* mRNA stability to inhibit autophagy in the progression of osteoarthritis[45]. In our study, we identified *AjULK* as a key gene for m6A-mediated autophagy of coelomocytes to resist *V. splendidus* infection. Further studies proved that the lack of AjULK abrogated the regulated m6A-mediated *V. splendidus*-induced autophagy. Our previous study confirmed that AjULK is an initiation factor of autophagy[24]. Thus, our results confirm that m6A mediates autophagy initiation via AjULK in *A. japonicus*.

In *Homo sapiens*, researchers also discovered that *ULK* transcripts are modified by m6A, and a lower m6A level of *ULK* is beneficial to promote its expression[47]. Interestingly, our study found a contrary phenomenon that AjMETTL3 increased the m6A level of *AjULK* to promote its expression in *A. japonicus*. A similar phenomenon, in which m6A-modifated transcripts had different fates in various species or biological processes, occurred in ATG5. In the seminoma of *H. sapiens*, upregulation of METTL3 can increase the m6A level of *ATG5* transcript, which increases the expression of ATG5[48]. However, METTL3 negatively regulates the expression of *ATG5* mRNA in an m6A-dependent manner in porcine embryos[49]. In addition, FTO decreases the m6A level of *ATG5* to promote the expression of ATG5 in adipogenesis[22]. This finding can be explained by the recognition of m6A-modified transcripts by different reader proteins, which determine the various fates of modified transcripts[50,51]. In *Homo sapiens*, m6A-modified *ULK* was recognized by YTHDF2, which accelerates the degradation of *ULK* mRNAs. However, our study showed that m6A-modified *AjULK* transcripts in *A. japonicus* were recognized by AjYTHDF, which presented the highest homology with YTHDF1. A number of studies have reported that the main function of YTHDF1 is to promote the expression of m6A-modified mRNA, and it has the opposite biological function to YTHDF2[30]. YTHDF1 can promote the translation efficiency of

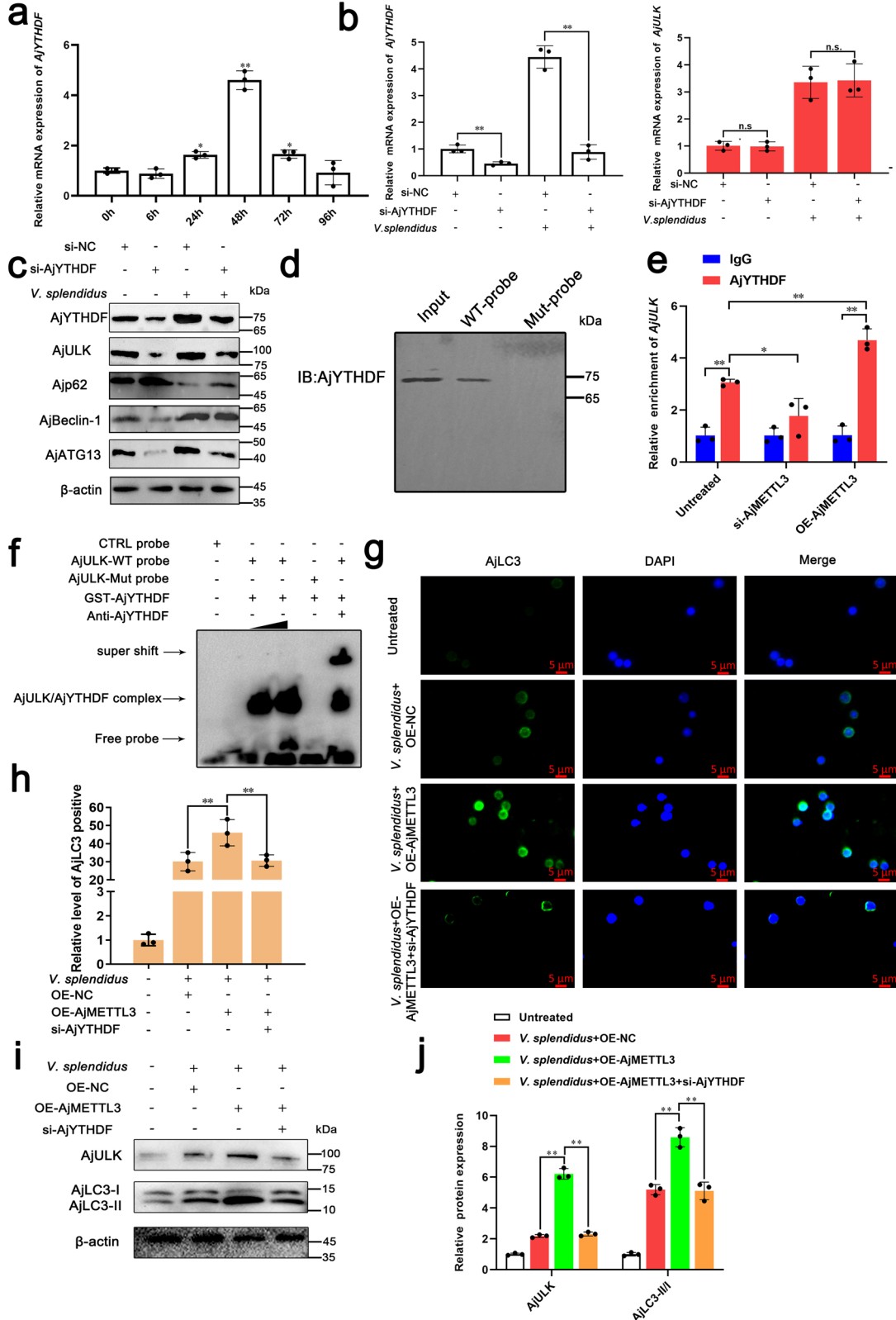

m6A-modified *Traf6* transcript to modulate the intestinal immune response to bacterial infection[52]. In the vascular remodeling of pulmonary hypertension, YTHDF1 promoted pulmonary hypertension by enhancing melanoma-associated antigen D1 expression[53]. In our studies, AjYTHDF can promote m6A-modified *AjULK* expression to mediate autophagy, which indicates a similar function with YTHDF1 in higher animals.

YTHDF1 contains an m6A-binding pocket in the YTH domain, in which K395 and Y397 are the key sites for YTHDF1 bound to the mRNA of m6A methylation[32]. Mutations in K395 and Y397 can abrogate the binding capacity of YTHDF1 with mRNA[54]. In AjYTHDF, similar sites were found at K464 and Y466. When the two sites are mutated, the capability of AjYTHDF to bind with *AjULK* mRNA and promote AjULK expression is eliminated.

**Fig. 4 AjYTHDF is a reader that mediates AjULK expression. a** qRT-PCR was used to analyze the expression pattern of *AjYTHDF* in coelomocytes during $1 \times 10^7$ CFU/mL *V. splendidus* infection of *A. japonicus*. The data are presented as means ± SDs ($n = 3$) relative to the 0 h and shown in bar graphs. Asterisks indicate significant differences compared with those at 0 h: *$p < 0.05$ and **$p < 0.01$ by one-way ANOVA. **b, c** After *A. japonicus* were infected or uninfected by $1 \times 10^7$ CFU/mL *V. splendidus* for 24 h in AjYTHDF knockdown condition, qRT-PCR and Western blot analysis of mRNA and protein levels of *AjYTHDF* and autophagy-related proteins in coelomocytes, respectively, were conducted. The protein levels of AjULK in the si-AjYTHDF group were significantly down-regulated compared with those of the proteins in the si-NC group. The mRNA level of *AjULK* in the si-AjYTHDF group did not change compared with that in the si-NC group. Under *V. splendidus* infection conditions, a similar phenomenon was observed. The data are presented as means ± SDs ($n = 3$) relative to the si-NC group and shown in bar graphs. Asterisks indicate significant differences between groups: *$p < 0.05$ and **$p < 0.01$ by one-way ANOVA. n.s., not significant ($p > 0.05$). The bands presented were from six blot membranes and same quantity of protein (50 μg) were loaded into each lane. **d** Immunoblot analysis of AjYTHDF after RNA pull-down assay showed its specific association with m6A-modified AjULK probe. **e** RIP qPCR analysis of the interaction between AjYTHDF and *AjULK* transcript in coelomocytes in AjMETTL3 knockdown or overexpression condition. When we increased the m6A level of *AjULK* through AjMETTL3 overexpression, the *AjULK* mRNA enriched by AjYTHDF increased. Conversely, down-regulated m6A level of *AjULK* by AjMETTL3 knockdown led to the interaction between AjYTHDF, and *AjULK* mRNA level was reduced. The data are presented as means ± SDs ($n = 3$) relative to the IgG group and shown in bar graphs. Asterisks indicate significant differences between groups: *$p < 0.05$ and **$p < 0.01$ by two-way ANOVA. **f** RNA EMSA showed the binding capability of purified GST-AjYTHDF with m6A-labeled AjULK. **g** After *A. japonicus* were infected by *V. splendidus* in AjMETTL3 overexpression and AjYHTHDF knockdown condition, immunofluorescence analysis of the fluorescence intensity of AjLC3 in coelomocytes revealed that knockdown of AjYTHDF abrogated the increased AjLC3 fluorescence signal by AjMETTL3 overexpression. Blue, green, and red signals represent nuclear, AjLC3, and lysosomes, respectively. Scale bar = 5 μm. **h** Relative AjLC3 positivity in 1000 cells from each indicated sample was determined. The data are presented as means ± SDs ($n = 3$) relative to the untreated group and shown in bar graphs. Asterisks indicate significant differences between groups: *$p < 0.05$ and **$p < 0.01$ by one-way ANOVA. **i** After *A. japonicus* were infected by *V. splendidus* in AjMETTL3 overexpression and AjYHTHDF knockdown condition, Western blot analysis detected that knockdown of AjYTHDF eliminated the up-regulated AjULK expression and AjLC3-II/I ratio by AjMETTL3 overexpression. The bands presented were from same blot membrane and same quantity of protein (50 μg) were loaded into each lane. **j** Protein band density was calculated using ImageJ software. The data are presented as means ± SDs ($n = 3$) relative to the untreated group and shown in bar graphs. Asterisks indicate significant differences between groups: *$p < 0.05$ and **$p < 0.01$ by one-way ANOVA. Error bars represent SDs.

---

The result illustrates that AjYTHDF mediated AjULK in an m6A-dependent manner, which is consistent with the YTHDF1 of other species. In mammals, YTHDF1 regulates mRNA translation by recruiting translation initiation factor complex 3 (eIF3), which promotes the rate-limiting step of translation for m6A-modified mRNAs by increasing their association with polysomes[55,56]. However, we did not detect eIF3 in the pull-down proteins of AjYTHDF. Notably, AjYTHDF can interact with AjEEF-1α, which is essential for m6A to promote AjULK expression. EEF-1α catalyzes the GTP-dependent binding of aminoacyl-transfer RNA (aa-tRNA) to ribosomes, which regulates the fidelity and rate of polypeptide elongation during translation[57]. In breast cancer lung metastasis and epithelial–mesenchymal transition, researchers discovered that YTHDF interacts with EEF to promote the expressions of m6A-modified transcripts, similar to our results[58,59]. Therefore, we speculated that AjYTHDF recognized and bound with m6A-modified site of *AjULK*, which recruited AjEEF-1α to accelerate aa-tRNA delivery to promote AjULK expression. However, how the complex regulates AjULK expression requires investigation in future studies.

Our study highlighted the importance of the m6A modification machinery in *A. japonicus* immunity and provided insights into the molecular mechanisms underlying *A. japonicus* autophagy. However, given the limitation of cell line in *A. japonicus*, experiments on vector expression (mRFP-GFP-LC3 assay, dual-luciferase assay, et al.) are not performed or only performed with the help of higher animal cell lines. In addition, the structure and number of autophagosomes in *A. japonicus* may differ from those of higher animals. Therefore, only an increase in autophagic fluorescence signal was observed in the immunofluorescence experiments, and no autophagic fluorescent spot signal was detected. In summary, the role of m6A and autophagy in *A. japonicus* immunity still needs to be explored in the future.

## Methods

### Experimental animals and ethics statement. The *A. japonicus* (98 ± 11 g) used in the study were purchased from Dalian Pacific Aquaculture Company and temporarily cultured in seawater (salinity 28 ± 0.5, temperature 16 ± 0.5 °C; pH 8 ± 0.1) for 3 days before the experiments. All experiments were performed in accordance with Guide for the Care and Use of Laboratory Animals of the National Institutes of Health. The experimental protocols were approved by the Research Animal Ethics Committee of Ningbo University, China.

### Challenge experiments. The challenge experiments were performed in accordance with the method of Lv et al.[60]. *V. splendidus* was cultured to optical density (OD) = 1 in 2216E culture medium (temperature, 28 °C; oscillation velocity, 180 r/min). The culture was centrifuged at 3000 g for 10 min to harvest *V. splendidus*. For challenge experiments, *A. japonicus* was divided into three groups, and each group contained three *A. japonicus*. Each challenge group was infected by immersion in a final concentration of $1 \times 10^7$ CFU/mL *V. splendidus*. For inhibition treatment, half of *A. japonicus* were injected with 10 nM Bafilomycin A1 (Baf-A1). The group without any treatment served as the untreated group. After 24 h treatment, coelomic fluids were obtained by dissecting the cavities of *A. japonicus* using sterilized scissors. First, the coelomic fluids were further filtered with 200-mesh filters and centrifuged at 800 g for 10 min to collect coelomocytes. Then, the collected coelomocytes were washed twice with sterilized isotonic buffer (0.001 M ethylenebis tetraacetic acid, 0.53 M NaCl, and 0.01 M Tris-HCl, pH 7.6). Finally, the coelomocytes were used for the detection of the corresponding RNA and protein levels in transmission electron microscopy (TEM) and immunofluorescence analyses.

### RNA interference. RNA interference experiments were performed in accordance with the method of Sun et al.[61]. The specific siRNAs for *AjMETTL3*, *AjULK*, *AjYTHDF*, and *AjEEF-1α* were synthesized by GenePharma and diluted to 20 μM with RNase-free $H_2O$ as the stock solution (Supplementary Table 1). Negative control (si-NC) was not targeted for any annotated genes in the *A. japonicus* transcriptome. For RNA interference, 10 μl siRNA, 10 transfection reagent (Beyotime Biotechnology, Beijing), and 80 μl phosphate-buffered saline (PBS) were mixed for 15 min to prepare a working solution. Every *A. japonicus* was injected with 100 μl working solution. After interference for 24 h, the coelomocytes were harvested for quantitative reverse transcription polymerase chain reaction (qRT-PCR), Western blot, TEM analysis, and immunofluorescence analysis. Each group included three replicates.

### RNA overexpression. RNA overexpression assay was performed following the method of Yang et al. and Chen et al.[62,63]. In brief, the *AjMETTL3*, *AjYTHDF* and *AjEEF-1α* open reading frame sequence (NCBI Accession: *AjMETTL3*: OQ863733; *AjYTHDF*: OQ401099, *AjEEF-1α*: OQ401098) were amplified with the primers in Supplementary Table 1 and inserted into pET-28a$^+$ containing a T7 promoter. Thereafter, the recombinant plasmid was used as a template to transcribe the single-stranded and capped *AjMETTL3* and *AjYTHDF* mRNA using T7 High-Efficiency Transcription Kit (Transgene, Beijing), in accordance with the manual instructions. The His tag mRNA from empty pET-28a$^+$ plasmid was used as overexpression-negative control (OE-NC). Each *A. japonicus* was injected with 300 μg mRNA. The overexpression efficiency was detected by qRT-PCR and Western blot 24 h post-injection. Afterward, *V. splendidus* were added to the final

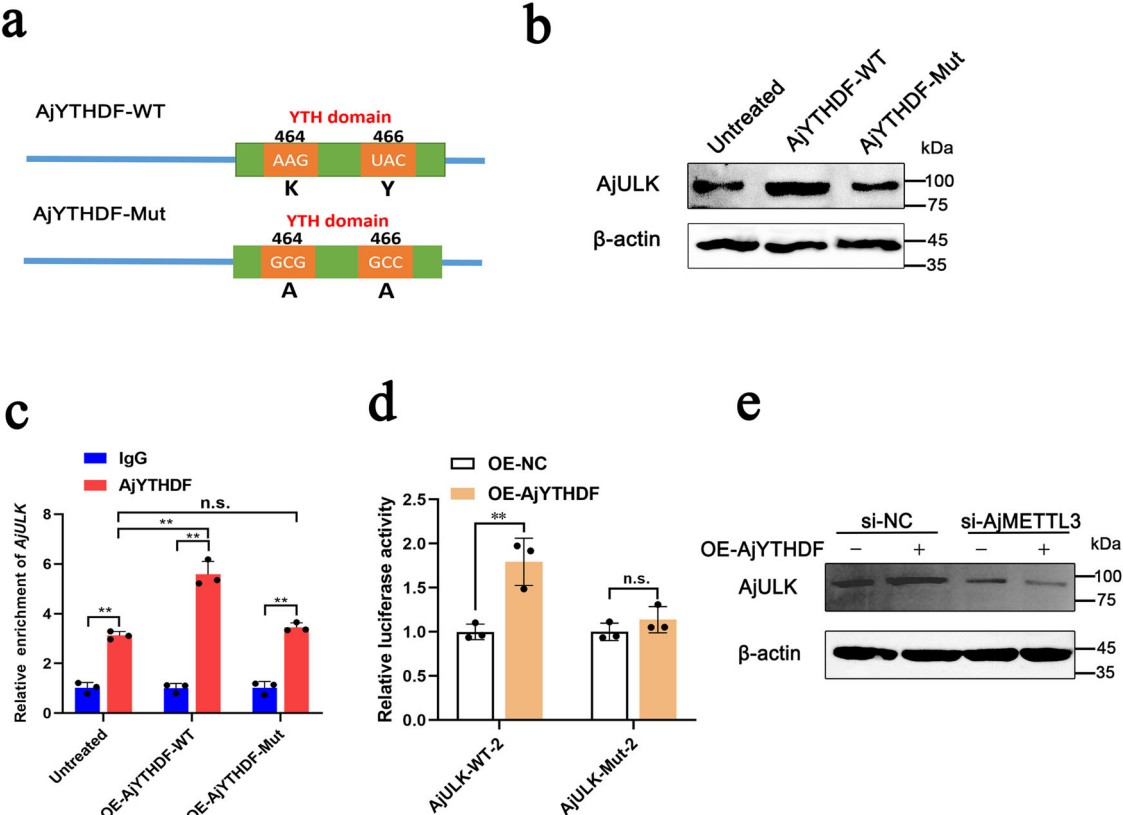

**Fig. 5 AjYTHDF regulates AjULK translation in an m6A-dependent manner. a** Schematic representation of AjYTHDF-WT and AjYTHDF-Mut constructs. **b** Western blot analysis of the protein level of AjULK in coelomocytes overexpressing AjYTHDF-WT or AjYTHDF-Mut. Upregulation of AjULK expression was observed in the AjYTHDF-WT group, whereas AjULK expression was unchanged in the AjYTHDF-Mut group. The bands presented were from two blot membranes and same quantity of protein (50 μg) were loaded into each lane. **c** After the overexpression of AjYTHDF-WT or AjYTHDF-Mut, RIP qPCR analysis showed the interaction between AjYTHDF and *AjULK* transcripts in coelomocytes. The enrichment of *AjULK* mRNA by AjYTHDF did not increase in the AjYTHDF-Mut group compared with the untreated group. The data are presented as means ± SDs ($n = 3$) relative to the IgG group and shown in bar graphs. Asterisks indicate significant differences between groups: $*p < 0.05$ and $**p < 0.01$ by two-way ANOVA. n.s., not significant ($p > 0.05$). **d** Relative luciferase activity of AjULK-WT-2 or AjULK-Mut-2 luciferase reporter AjYTHDF overexpressing cells. Overexpression of AjYTHDF significantly up-regulated luciferase reporter activity in the AjULK-WT-2 group, whereas the luciferase reporter activity of AjULK-MUT-2 showed no change. Renilla luciferase activity was measured and normalized to firefly luciferase activity. The data are presented as means ± SDs ($n = 3$) relative to the OE-NC group and shown in bar graphs. Asterisks indicate significant differences between groups: $*p < 0.05$ and $**p < 0.01$ by two-way ANOVA. **e** Western blot analysis of AjULK expression in coelomocytes of overexpression of AjYTHDF together with either si-NC or si-AjMETTL3. Overexpression of AjYTHDF cannot upregulate AjULK expression in coelomocytes lacking AjMETTL3. The bands presented were from two blot membranes and same quantity of protein (50 μg) were loaded into each lane. Error bars represent SDs.

concentration of $10^7$ CFU/mL and incubated for another 24 h. Finally, the coelomocytes were harvested by the above-mentioned method for further research. The experiments were repeated thrice.

**TEM**. TEM was conducted in accordance with the method of Meng et al.[64]. The coelomocytes were obtained following the above-mentioned method and centrifuged to obtain pellets. The pellets were then fixed in 2.5% glutaraldehyde in PBS at 4 °C for 2 h and washed once with 0.1 M PBS. Subsequently, the pellets were fixed in 1% osmium tetroxide for 1.5 h, dehydrated through a series of ethanol concentrations, and embedded in Epon resin. The samples were sectioned with a microtome, and the sections were double stained with 3% uranyl acetate and lead citrate before examination under a transmission electron microscope (HITACHI, Japan).

**Immunofluorescence analysis**. Immunofluorescence analysis was performed referring to the method of Zhao et al.[65]. The coelomocytes were resuspended as cell suspension with L-15 medium and then added in a cell culture plate containing glass chamber slides at a density of approximately $10^5$ cells ml$^{-1}$ for 6 h at 16 °C. Thereafter, the L-15 medium was removed, and the glass chamber slides were washed twice with PBS. Subsequently, the coelomocytes on glass chamber slides were fixed with 4% paraformaldehyde for 30 min and permeabilized with 0.1% Triton X-100 (Sigma, USA) for 10 min. Further, the coelomocytes were washed thrice with PBS and blocked with 5% bovine serum albumin (BSA) at room

temperature for 2 h. After the blocking buffer was removed, the coelomocytes were incubated overnight with primary antibody (anti-LC3 antibody, anti-lysosome associated membrane protein (LAMP) antibody, 1: 500 dilution) at 4 °C. After washing for three times with PBST, the coelomocytes were incubated with secondary antibody (fluorescein isothiocyanate-conjugated goat anti-mouse and Cy3-conjugated goat anti-rabbit, 1:1000 dilution) at room temperature for 1.5 h. The coelomocytes were washed thrice with PBST, and the nuclei were stained with 4',6-diamidino-2-phenylindole (1:10,000 dilution, Beyotime Biotechnology, Beijing). After the final three cycles of washing, the cells were sealed with antifade fluorescence mounting medium and observed with a laser-scanning confocal microscope (ZEISS, Germany).

**RNA isolation and qRT-PCR**. The coelomocytes collected were dissolved, and RNA was isolated with RNAiso Plus (TaKaRa, Japanese). The isolated RNA was synthesized to cDNA using PrimeScript™ RT reagent Kit with gDNA Eraser (TaKaRa, Japanese). The relative mRNA expression of each gene was analyzed using an Applied Biosystem 7500 Real-time Quantitative PCR System (Thermo Fisher Scientific, USA) and TB Green® Fast qPCR Mix (TaKaRa, Japanese). Each reaction contained a reaction volume of 20 μl, which included 8 μl cDNA (1: 50 dilution), 0.8 μl each primer (10 μM), 0.4 μl ROXII, and 10 μl TB Green PCR Master Mix (TaKaRa, Japanese). The amplification procedure was as follows: denaturation at 95 °C for 1 min followed by 40 cycles of 95 °C for 5 s and 60 °C for 34 s. After the cycling stage, melting curve analyses were performed. Each group

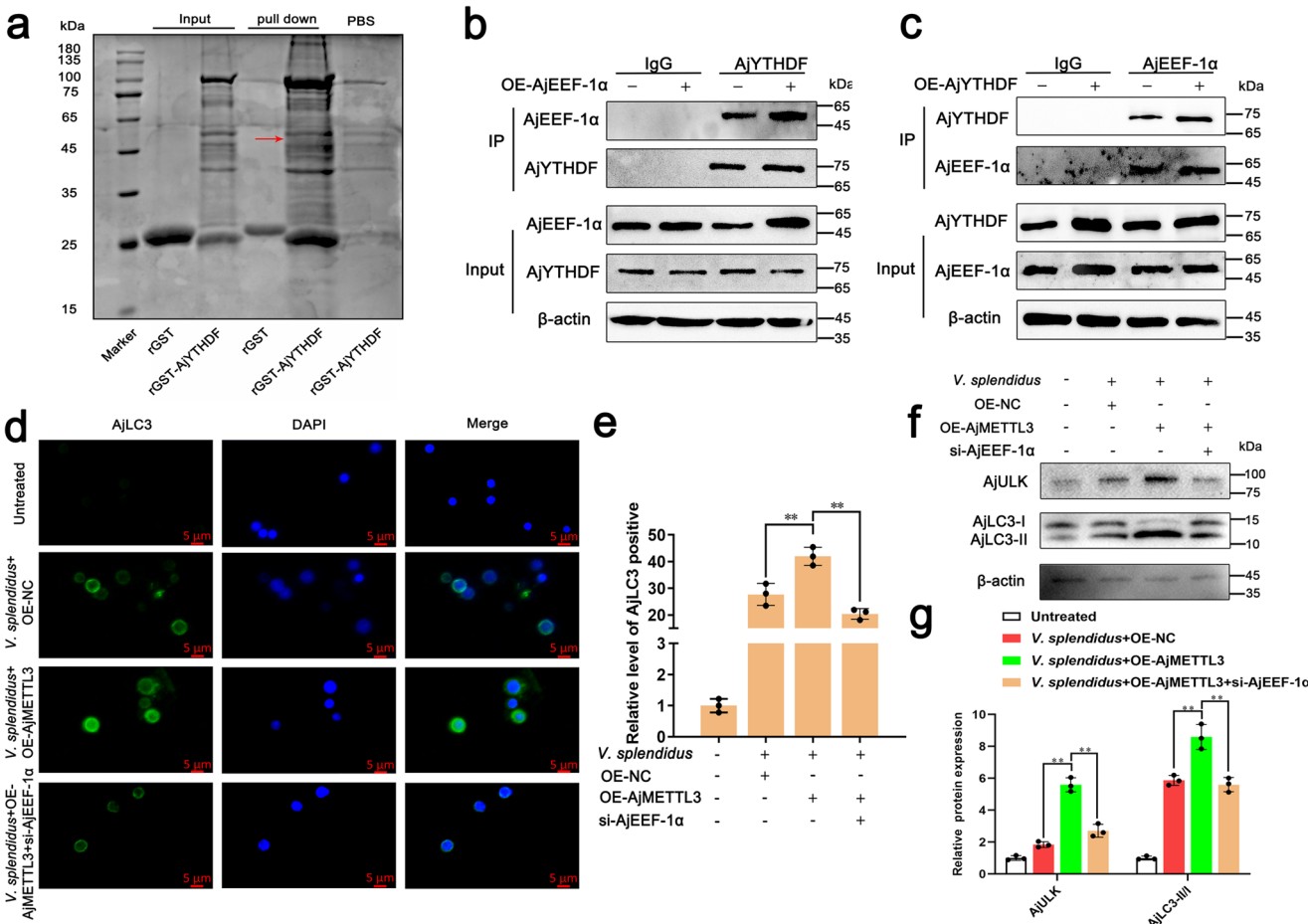

**Fig. 6 AjYTHDF combines with AjEEF-1α to promote AjULK expression. a** GST pull-down assay identified the interacting proteins of AjYTHDF from coelomocytes. **b, c** co-IP experiment confirmed the interaction between AjYTHDF and AjEEF-1α in coelomocytes. The bands presented in the two figures were from five blot membranes and same quantity of protein (50 μg) were loaded into each lane. **d** After *A. japonicus* were infected by *V. splendidus* in AjMETTL3 overexpression and AjEEF-1α knockdown conditions, immunofluorescence analysis of fluorescence intensity of AjLC3 in coelomocytes and knocked down AjEEF-1α eliminated the increased fluorescence intensity of AjLC3 in AjMETTL3 overexpression condition. Blue and green signals represent nuclear and AjLC3, respectively. Scale bar = 5 μm. **e** Relative AjLC3 positivity in 1000 cells from each indicated sample was determined. The data are presented as means ± SDs (*n* = 3) relative to the untreated group and shown in bar graphs. Asterisks indicate significant differences between groups: *$p < 0.05$ and **$p < 0.01$ by one-way ANOVA. **f** After *A. japonicus* were infected by *V. splendidus* in AjMETTL3 overexpression and AjEEF-1α knockdown conditions, Western blot analysis of AjULK expression and AjLC3-II/I ratio in coelomocytes. Knocked down AjEEF-1α eliminated the AjULK, and AjLC3-II/I ratio increased in AjMETTL3 overexpression condition. The bands presented were from same blot membrane and same quantity of protein (50 μg) were loaded into each lane. **g** Protein band density was calculated using ImageJ software. The data are presented as means ± SDs (*n* = 3) relative to the untreated group and are shown in bar graphs. Asterisks indicate significant differences between groups: *$p < 0.05$ and **$p < 0.01$ by one-way ANOVA. Error bars represent SDs.

trial was run in triplicate parallel reactions and repeated thrice. The primers are listed in Supplementary Table 1. The $2^{-\Delta\Delta CT}$ method was used to analyze the expression level of each gene. The full equation for $2^{-\Delta\Delta CT}$ method as follow:

$$\Delta CT = CT_{target} - CT_{reference} \tag{1}$$

$$\Delta\Delta CT = \left(Sample\ CT_{target} - Sample\ CT_{reference}\right) \\ - \left(control\ CT_{target} - control\ CT_{reference}\right) \tag{2}$$

$$2^{-\Delta\Delta CT} = 2^{-\left[\left(Sample\ CT_{target} - Sample\ CT_{reference}\right) - \left(control\ CT_{target} - control\ CT_{reference}\right)\right]} \tag{3}$$

**Western blot analysis**. Protein extracts from coelomocytes or immunoprecipitation samples were prepared using cell lysis buffer (Beyotime Biotechnology, Beijing). The concentration was measured with a BCA protein assay kit (Cwbio, Beijing). The total proteins (50 μg) of each sample were separated by sodium dodecyl sulphate-polyacrylamide gel electrophoresis (SDS-PAGE) and then transferred to 0.22 μm polyvinylidene difluoride membrane (Millipore, USA). The membranes were blocked with 5% skim milk at room temperature for 2 h. Then, the membranes were incubated with primary antibodies (1:500 dilution in 5% BSA

solution) at 4 °C overnight. Thereafter, the membranes were washed thrice with TBST (20 mM Tris-HCl, 150 mM NaCl, and 0.05% Tween-20; pH = 7.6) and incubated with secondary antibodies (1:5000 dilution in 5% BSA solution) at room temperature for 1.5 h. Finally, the membranes were washed four times, incubated with Western Lightning ECL substrate (Bio-Rad, USA), and imaged by Omega Lum C imaging system (Aplegen, USA). Supplementary Table 2 shows the antibodies used in this study.

**Methylated RNA (MeRIP)-qPCR**. MeRIP-qPCR assay was performed using EpiQuik CUT&RUN m6A RNA Enrichment Kit (Epigentek, USA) in accordance with the manufacturer's instructions. Firstly, we isolated 300 μg total RNA from coelomocytes of each sample. Next, 30 μg total RNA was saved as the input control and the others were used for immunoprecipitation. Protein A/G magnetic beads were prewashed three times with wash buffer and then incubated with 5 μg anti-m6A antibody (1:10 dilution) or rabbit IgG for 2 h at 4 °C with shake. Next, the protein A/G magnetic beads with m6A antibody were washed for three times using wash buffer and incubated in immunoprecipitation buffer with total RNA for 2 h at 4 °C. Next, the protein A/G magnetic beads with m6A antibody and methylated mRNAs were washed three times with wash buffer and eluted by elution buffer. The mRNAs in elution buffer were precipitated with 5 mg glycogen and one-tenth volumes of 3 M sodium acetate in a 2.5 volume of 100% ethanol for 8 h at −80 °C.

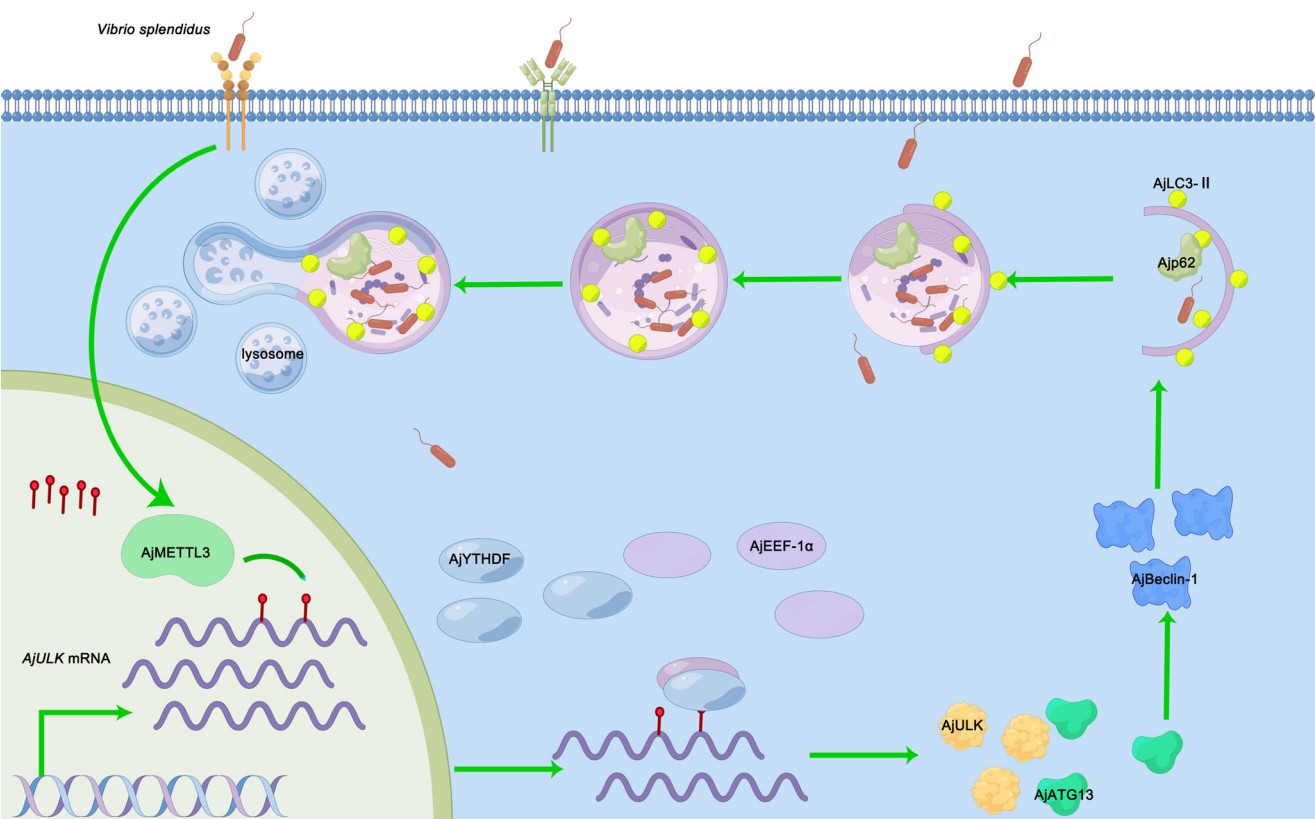

**Fig. 7 Proposed mechanism of m6A-modulated autophagy in *A. japonicus* in response to *V. splendidus* infection.** Upon *V. splendidus* infection, the mRNA of autophagy initial factor *AjULK* was methylated by AjMETTL3 in the nucleus and then transported to the cytoplasm, where m6A-modified *AjULK* was recognized and bound with conserved reader protein AjYTHDF. Subsequently, AjYTHDF interacted with AjEEF-1α to recruit aa-tRNA to promote AjULK protein expression, which activated the autophagy to degrade the intracellular *V. splendidus*.

Finally, the precipitated RNA was calculated by qRT-PCR, and the corresponding m6A enrichment in each sample was calculated by normalizing to the input.

**Site mutation**. Site mutation was conducted using Site-Directed Mutagenesis Kit (Sangon Biotech, Shanghai) in accordance with the manufacturer's instructions. First, two complementary mutagenic primers with centrally located mutation sites were synthesized (Supplementary Table 1). Next, 5 ng vector, which was inserted into the open reading frame sequence of *AjULK* (NCBI Accession: MH807452) and *AjYTHDF*, was used as a template, and the entire target vector was amplified with the primers. Then, PCR production used D*pn* I restriction endonuclease to digest the residual methylated template and any hemimethylated DNA. Finally, 10 μl PCR products with the desired mutation site were transformed into competent cell DH5α, which was sequenced, and the plasmid was extracted for subsequent experiments.

**Dual-luciferase assay**. The dual-luciferase assay was performed referring to the method of Song et al.[66]. The m6A sequences of *AjULK* and its mutant were ligated into the psiCHECK-2 vector. The mutant sequences were obtained by the Fast Mutagenesis System (Transgene, Beijing). Meanwhile, the sequences of *AjMETTL3* and *AjYTHDF* were ligated into pCMV-Flag 2 C. After epithelioma papulosum cyprinid cells (Zebrafish Resource Center, Wuhan) were cultured on 96-well plates at 28 °C overnight, the recombinant plasmids (200 ng per well) were transfected into the epithelioma papulosum cyprinid cells with Lipofectamine 6000 (Beyotime Biotechnology, Beijing). The empty pCMV-Flag 2 C was used as the OE-NC. After transfection for 48 h, the culture medium was removed, and each well was washed twice with 100 μl PBS. Next, each well was incubated with 20 μl 1× Passive Lysis Buffer for 30 min at room temperature. Then, 100 μl Luciferase Assay Reagent II was added to each well, and the firefly luciferase activity was immediately measured by a GloMax 96, 20/20 luminometer (Promega, USA). After measuring the firefly luciferase activity, 100 μl Stop & Glo® Reagent was added accordingly to measure the Renilla luciferase activity. All group data came from three biological replicates.

**RNA pull down**. Biotin-labeled *AjULK* RNA containing methylated or unmethylated adenosine was synthesized by Sangon Biotech (Shanghai, China). RNA pull-down assays were carried out as described in the Pierce Magnetic RNA-Protein Pull Down Kit (Thermo Scientific, USA). Briefly, 50 pmol RNA was incubated with 50 μl streptavidin-coated magnetic beads (Sangon Biotech, Shanghai) overnight at

4 °C. The fresh coelomocyte lysates were added to the binding reaction with RNase inhibitor and protease/phosphatase inhibitor cocktail for 1 h. Then, the bound proteins were eluted from the packed beads and analyzed by Western blot analysis. The probe sequences are shown in Supplementary Table 2.

**RNA immunoprecipitation (RIP)**. RIP was conducted using the RNA Immuno-precipitation Kit (Geneseed, Guangzhou) in accordance with the manufacturer's instructions. In brief, the collected coelomocytes were lysed in 1 mL RIP lysis buffer. A total of 100 μl cell lysates were saved as the input control. Then, magnetic beads coated with 10 μg specific antibodies against IgG and AjYTHDF were incubated with the prepared cell lysates at 4 °C overnight. Thereafter, the RNA-protein complexes were washed thrice and incubated with proteinase K digestion buffer to isolate the immunoprecipitated RNA. The relative interaction between *AjULK* mRNA and protein was determined by qRT-PCR.

**RNA electrophoretic mobility shift assay (EMSA)**. We synthesized biotin-labeled *AjULK* RNA containing methylated or unmethylated adenosine from Sangon Biotech (Shanghai, China). (Shanghai, China). RNA EMSA assay was performed by LightShift™ Chemiluminescent RNA EMSA Kit (Thermo Fisher Scientific, USA) according to the manufacturer's instructions. First, biotin-labeled oligonucleotide probe (*AjULK* RNA containing methylated or unmethylated adenosine) and purified AjYTHDF proteins (500 ng) were mixed and incubated in binding buffer at room temperature for 20 min. For super shift assays, purified AjYTHDF proteins were preincubated with AjYTHDF antibodies on ice for 20 min and then incubated with the biotin-labeled probe at room temperature for 20 min. Next, the RNA-protein complexes were mixed with 5× loading buffer and separated by 6% nondenaturing polyacrylamide gel in 0.5 × Tris-borate-EDTA buffer at 100 V for 1 h. Then, the proteins and RNA in the gel were transferred to a nylon membrane (Solarbio, Beijing) in 0.5 × Tris-borate-EDTA buffer at 400 mA for 30 min. Next, the membrane was crosslinked by ultraviolet (254 nm) for 15 min and then blocked in blocking buffer at room temperature for 30 min. Next, the membrane was incubated in horseradish peroxidase-linked streptavidin (1:200 dilution) at room temperature for 30 min. Next, the membranes were incubated with ECL substrate (Bio-Rad, USA) and imaged by Omega Lum C imaging system (Aplegen, USA). The probe sequences are shown in Supplementary Table 1.

**Glutathione-S-transferase (GST) pull-down assay**. GST pull-down assay was performed in accordance with the method of Gao et al.[67]. This type of assay was conducted to filter AjYTHDF-interacting proteins. The *AjYTHDF* open reading frame sequence were inserted into pgex4T-2 vector, which was transformed into *E. coli* Rosseta to express the GST-tagged AjYTHDF protein (GST-AjYTHDF). GST-AjYTHDF was immobilized by affinity chromatography using GST Sefinose™ Resin (Sangon Biotech, Shanghai). Then, 500 μg total protein from *A. japonicus* coelomocytes was added to the prepared GST Sefinose™ Resin with GST-AjYTHDF and incubated overnight at 4 °C. Next, the resin was washed thrice with GST wash buffer (Sangon Biotech, Shanghai) to remove protein impurities. Finally, the resin was eluted with GST elution buffer (Sangon Biotech, Shanghai) to obtain the protein complexes, which were analyzed by SDS-PAGE and liquid chromatography-tandem mass spectrometry (MS).

**Protein co-immunoprecipitation (co-IP)**. Co-IP was performed referring to the method of Ma et al.[68]. A total of 20 μl protein A + G beads (Beyotime Biotechnology, Beijing) were washed thrice with TBS (20 mM Tris-HCl, 150 mM NaCl, and pH = 7.6) and incubated with 500 μl antibody working solution (50 μg/mL) at 4 °C for 1 h. The coelomocytes from different treatment groups were lysed on ice with RIPA lysis buffer (Beyotime Biotechnology, Beijing) for 30 min. Then, 500 μl supernatant and the prepared protein A + G beads were incubated together at 4 °C overnight. Next, the complexes were washed thrice using TBS and eluted by 1 × SDS-PAGE buffer for Western blot assay.

**Intracellular bacterial burden assay**. Intracellular bacterial abundance assay was performed referring to method of Li et al. and Liu et al.[69,70]. In brief, 100 μl *V. splendidus* ($10^6$ CFU/mL) was injected into different *A. japonicus* groups, including the untreated, si-NC, si-AjMETTL3, OE-AjMETTL3, OE-NC, and OE-AjMETTL3 + si-AjULK groups. After infection for 12 h, 10,000 coelomocytes were collected from each group and washed thrice by PBS. Then, the coelomocytes were resuspended by 1 ml PBS. Next, 10 μl diluted fluid from each group was evenly spread onto the 2216E agar plates and cultured at 28 °C. After 24 h, the bacteria were quantified using the plate counting method. The experiment was performed with three independent replicates.

**Statistics and reproducibility**. For all the statistical analyses and production of the graphs GraphPad Prism version 8 was used. The data are presented as the means ± SDs (n = 3) relative to the untreated group and shown in bar graphs. Error bar represent SDs. Data points on graphs represent three biological replicates. Student's t-test was used to determine the significance levels between two samples. One-way or two-way ANOVA was used to determine significance levels between three or more treatments.

**Reporting summary**. Further information on research design is available in the Nature Portfolio Reporting Summary linked to this article.

## Data availability
The data that support the findings of this study are available in the methods and/or supplementary material of this article. NCBI Accession: *AjMETTL3*: OQ863733; *AjULK*: MH807452; *AjYTHDF*: OQ401099, *AjEEF-1α*: OQ401098. The m6A-seq results had been uploaded to online repositories with accession number of PRJNA774950. The Fig. 7 was constructed by FigDraw (www.figdraw.com/, ID: OTARUffa00). Uncropped Western blot images are provided in Supplementary Figure 5–19. All statistical source data that underlie the graphs in figures are provided in Supplementary Data 1.

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

## Acknowledgements
This work was financially supported by National Natural Science Foundation of China (32073003, 42176102), Natural Science Foundation of Zhejiang Province (LZ19C190001), and the K.C. Wong Magna Fund in Ningbo University.

## Author contributions
J.Q.L., Y.N.S. and C.H.L. conceived and designed the project. J.Q.L. conducted all the experiments and wrote the first draft. Y.N.S., D.D.L. and C.H.L revised the manuscript. All authors read and approved the final version of the manuscript.

## Competing interests
The authors declare no competing interests.
