## [Peer Review File · Communications Biology]

Reviewers' comments:

Reviewer #1 (Remarks to the Author):

This study by Jiqing Liu and colleagues identified m6A RNA methylation as a key regulator of xenophagy in echinoderms. They found that inhibition of m6A levels in *Apostichopus japonicus* suppresses *Vibrio splendidus*-induced coelomocyte autophagy, leading to an increase in the intracellular *V. splendidus* burden. They further identified a signaling pathway, AjULK-AjYTHDF/AjEEF-1 α , which regulates xenophagy in echinoderms. Overall, the manuscript is well organized, and the data are presented in a clear and logical way. The manuscript only requires couples of improvements. Major comments:

1. English editing will be needed to improve the manuscript. I found grammar/spelling mistakes across the manuscript.
2. In Figure 2a-b, does data come from MeRIP-seq result? I didn't see any m6A-seq-related description in material methods and references.
3. If they did MeRIP-seq, they should provide more info regarding seq results (like Go enrichment and m6A motif analysis), so we can judge whether the seq data is trustworthy.
4. Where are the m6A sites on AjULK? 3'UTR or intron? Are they conserved in different species?

Reviewer #2 (Remarks to the Author):

This study presents the first evidence that deficiency of AjMETTL3 in *Apostichopus japonicus* can significantly reduce *V. splendidus*-induced coelomocyte autophagy and increase the intracellular *V. splendidus* burden, through a mechanism related to AjYTHDF-mediated AjULK protein expression. However, there are several experimental and writing issues in this manuscript, which are outlined below.

Major points:

1. In all figure legends, except for Figure 7, the authors state that "asterisks indicate significant differences compared to controls," suggesting that the sample treated with or without *V. splendidus*, si-AjMETTL3, OE-AjMETTL3, and so on was only compared to the control group. This is inconsistent with the description in the results section, which compares the si-AjMETTL3 group to the si-NC group under *V. splendidus* infection conditions. These inconsistencies raise concerns about the reliability of the statistics, and we suggest that the authors perform a second statistical examination of the data for the entire article.
2. The authors mention that "In this condition, both the protein level of AjULK and downstream genes of AjULK were significantly decreased (Fig. 4c) (line377-379)" in the manuscript. However, the protein expression level of Ajp62 is elevated in Figure 4c. The authors should clarify this discrepancy.
3. The scale of the second diagram in Figure 3c does not match that of the other diagrams. Additionally, the textual annotation "control + - -" in the statistics charts of Figures 1c, 3d, 3e, 3f, 4e, and 4f is unclear and should be revised.
4. The authors' demonstration of the mechanism underlying the impact of AjYTHDF on the expression of the AjULK protein is insufficient. Additional experiments should be included to clarify this mechanism.
5. Figure 6c is not described in the RESULTS section of the manuscript and should be addressed.
6. The manuscript's writing could be improved. Some sentences, such as lines 303–307 and 382–384, are not clear enough, and some grammatical errors, like the one in line 32, should be corrected. Need to be professionally proofread.

Minor points:

1. The summary is too long with too much details.
2. The DISCUSSION section of the manuscript is wordy and should be edited for coherence. The authors should also better explain the limitations of their work in the final part of the Discussion.

In conclusion, we do not recommend this manuscript for publication in Communications Biology.

Dear Reviewers,

Thank you very much for your letter and the comments about our paper “N6-methyladenosine resists *Vibrio splendidus* infection by targeting coelomocyte autophagy via AjULK-AjYTHDF/AjEEF-1 α axis in echinoderms (COMMSBIO-23-0314), which helped improve our paper’s quality. We have considered the comments and corrected them carefully. We proofread our manuscript for mistakes and grammatical errors by using American Journal Experts. Here we submit a revised manuscript with color-coded as well as the detailed responses. We appreciate for your warm work earnestly, and hope that the revised manuscript will meet with approval. The following is our response to each comment point by point.

Reviewer #1

1. English editing will be needed to improve the manuscript. I found grammar/spelling mistakes across the manuscript.

Authors response: We are really sorry for the grammar/spelling mistakes of our manuscript and improve the language by using ShineWrite.com Editing Service with ID of 40-2023032912135865.

2. In Figure 2a-b, does data come from MeRIP-seq result? I didn’t see any m6A-seq-related description in material methods and references.

Authors response: We thank the reviewer for your nice reminder. The MeRIP-seq results had been uploaded to online repositories with accession number of PRJNA774950 (<https://www.ncbi.nlm.nih.gov/pmc/?term=PRJNA774950%20>). We have clarified this important information in our revised manuscript and also provided our previous published work (Line 355-356).

3. If they did MeRIP-seq, they should provide more info regarding seq results (like Go enrichment and m6A motif analysis), so we can judge whether the seq data is trustworthy.

Authors response: We appreciate the reviewer’s valuable and professional comments. These indicated MeRIP-seq results could be found in our previous work (Shao et al., Dev. Comp. Immunol. 2022, 133:104434). In the present work, we focused

on the m6A modified autophagy-related genes in *Apostichopus japonicus*, and 15 autophagy-related genes were detected from 6159 differentially expressed m6A modified genes (Shao et al., Dev. Comp. Immunol. 2022, 133:104434; Shao et al., J. Biol. Chem. 2022, 298:101667). Then, we used these genes to construct a heat map, and found that AjULK was the highest changed gene at m6A methylation levels. We have clarified this important point in our revised manuscript (Line 353-358).

4. Where are the m6A sites on AjULK? 3'UTR or intron? Are they conserved in different species?

Authors response: We thank the reviewer for this comment. In our work, the two m6A sites were identified from the CDS regions of AjULK mRNA, in which only one site could mediate AjULK expression by dual-luciferase assay (Fig. 2h). The detail information of AjULK m6A sites was added in our revised manuscript (Line 360-361). Moreover, the m6A sites on ULK transcripts were not conserved in different species based on reported studies. Researchers found three m6A sites were detected in the 3'-UTR of human ULK transcript (Jin et al, Cell Res. 2018, 28, 955-957). To further address this issue, we used SRAMP (<http://www.cuilab.cn/sramp>) to predict ULK m6A sites of other species, including *Xenopus tropicalis*, *Drosophila melanogaster*, *Bos taurus*, *Danio rerio* and *Asterias rubens*, which further supported that the m6A sites on ULK were not conserved in different species.

Reviewer #2

Major points:

1. In all figure legends, except for Figure 7, the authors state that "asterisks indicate significant differences compared to controls," suggesting that the sample treated with or without *V. splendidus*, si-AjMETTL3, OE-AjMETTL3, and so on was only compared to the control group. This is inconsistent with the description in the results section, which compares the si-AjMETTL3 group to the si-NC group under *V. splendidus* infection conditions. These inconsistencies raise concerns about the reliability of the statistics, and we suggest that the authors perform a second statistical examination of the data for the entire article.

Authors response: We appreciated the reviewer's constructive comments and have revised the figures accordingly. In the revised manuscript, we perform a second statistical examination of the data for the entire article and reconstructed all bar chart. The data were compared between the two groups with only one single variable.

2. The authors mention that "In this condition, both the protein level of AjULK and downstream genes of AjULK were significantly decreased (Fig. 4c) (line377-379)" in the manuscript. However, the protein expression level of Ajp62 is elevated in Figure 4c. The authors should clarify this discrepancy.

Authors response: We appreciate the reviewer's professional comments and revised the sentence to "Consistently, the expression of AjULK-downstream genes, including AjBeclin-1 and AjATG13, were also significantly depressed, whereas the protein level of Ajp62 was significantly elevated in this condition (Figs. 2e and 2f)" in our resubmitted manuscript. In the process of autophagosome formation, p62, as a bridge between LC3 and polyubiquitination protein, was selectively wrapped into the autophagosome, and then degraded by proteolytic enzymes in the autophagolysosome (Zhang et al. Hepatology 2019, 69, 1046-1063. Choi, et al. Nat. Commun. 2020, 11, 1386 (2020). Cosin-Roger et al. Nat. Commun. 2017, 8, 98). Therefore, the protein expression level of p62 was negatively correlated with autophagy activity. In my previous study, we also discovered the similar phenomenon that the expression level of Ajp62 protein was negatively correlated with autophagy activity in *Apostichopus japonicus* (Shao et al., J. Biol. Chem. 2022, 298:101667. Chen, Aquaculture 2021, 532, 736022).

3. The scale of the second diagram in Figure 3c does not match that of the other diagrams. Additionally, the textual annotation "control + - -" in the statistics charts of Figures 1c, 3d, 3e, 3f, 4e, and 4f is unclear and should be revised.

Authors response: We appreciate the reviewer's comment and have updated a new Figure 3c with the same scale to other images in our revised manuscript. Moreover, the "control + - -" was deleted from the revised manuscript, which was originally indicated untreated group.

4. The authors' demonstration of the mechanism underlying the impact of

AjYTHDF on the expression of the AjULK protein is insufficient. Additional experiments should be included to clarify this mechanism.

Authors response: We appreciate the reviewer's valuable comments. To demonstrate the mechanism underlying the impact of AjYTHDF on the expression of the AjULK protein, we performed RNA pull down and RNA EMSA assays to explore the binding between AjYTHDF1 and AjULK mRNA. The results confirmed that AjYTHDF could specifically bind with m6A-modified AjULK transcripts rather than unmodified AjULK. We have added detailed description about the results of RNA pull down and RNA EMSA in our revised manuscript (Figs. 4e and 4g).

5. Figure 6c is not described in the RESULTS section of the manuscript and should be addressed.

Authors response: We appreciate the reviewer's comment and add the results in our revised manuscript (Line 473-475).

6. The manuscript's writing could be improved. Some sentences, such as lines 303–307 and 382–384, are not clear enough, and some grammatical errors, like the one in line 32, should be corrected. Need to be professionally proofread.

Authors response: We thank the review for your careful proofreading and improve the language by using ShineWrite.com Editing Service with ID of 40-2023032912135865, especially the sentence you indicated.

Minor points:

1. The summary is too long with too much details.

Authors response: The summary sentence has been improved to meet with your comments. Special thanks for your suggestion.

2. The DISCUSSION section of the manuscript is wordy and should be edited for coherence. The authors should also better explain the limitations of their work in the final part of the Discussion.

Authors response: We are appreciative of the reviewer's suggestion and add the limitation of the work in the last paragraph. Moreover, the language and coherence of the section were further improved to our best.

We deeply appreciate your all the valuable comments and suggestions, and look forward to hearing from you regarding our submission. We would be glad to respond to any further questions and comments that you may have.

Sincerely,

Chenghua Li

818 Fenghua Road,

Ningbo University,

Ningbo, Zhejiang Province 315211, P. R. China

Email: lichenghua@nbu.edu.cn

REVIEWERS' COMMENTS:

Reviewer #1 (Remarks to the Author):

I have no more questions.

Reviewer #2 (Remarks to the Author):

I am satisfied with the revision.

Author response

Thank you very much for your letter and the comments about our paper “N6-methyladenosine helps *Apostichopus japonicus* resist *Vibrio splendidus* infection by targeting coelomocyte autophagy via the AjULK-AjYTHDF/AjEEF-1 α axis (COMMSBIO-23-0314), which helped improve our paper’s quality. The following is our response to each comment point by point.

Reviewer #1 (Remarks to the Author):

I have no more questions.

Authors response: Thanks very much for the reviewer’s positive comments.

Reviewer #2 (Remarks to the Author):

I am satisfied with the revision.

Authors response: Thanks very much for the reviewer’s positive comments.

Sincerely,

Chenghua Li

818 Fenghua Road,

Ningbo University,

Ningbo, Zhejiang Province 315211, P. R. China

Email: lichenghua@nbu.edu.cn